Identification of potential biomarkers of vascular calcification using bioinformatics analysis and validation in vivo

Chen Chuanzhen 1
Wu Yinteng 2
Lu Hai-lin 1
Liu Kai 1
Qin Xiao 1 dr_qinxiao@hotmail.com
1 Department of Vascular Surgery, The First Affiliated Hospital of Guangxi Medical University , Nanning, Guangxi Province , China
2 Department of Orthopedic and Trauma Surgery, The First Affiliated Hospital of Guangxi Medical University , Nanning, Guangxi Province , China
Albertini Maria Cristina
Electronic publication date: 2022 Mar 16
Publication date: 2022
Volume: 10
Electronic Location ID: e13138
Received 2021 Nov 29; Accepted 2022 Feb 28
Copyright: © 2022 Chen et al.
Copyright year: 2022
Copyright holder: Chen et al.
License: This is an open access article distributed under the terms of the Creative Commons Attribution License, which permits unrestricted use, distribution, reproduction and adaptation in any medium and for any purpose provided that it is properly attributed. For attribution, the original author(s), title, publication source (PeerJ) and either DOI or URL of the article must be cited.
License URL: https://creativecommons.org/licenses/by/4.0/

Keywords: Vascular calcification, Differentially expressed genes, Gene set enrichment analysis, Gene set variation analysis, Protein–protein interaction, Functional enrichment analysis

Funding: The National Natural Science Foundation of China 81960091 This work was supported by the National Natural Science Foundation of China (No: 81960091). The funders had no role in study design, data collection and analysis, decision to publish, or preparation of the manuscript.

==============================
Background

Vascular calcification (VC) is the most widespread pathological change in diseases of the vascular system. However, we know poorly about the molecular mechanisms and effective therapeutic approaches of VC.

Methods

The VC dataset, GSE146638, was downloaded from the Gene Expression Omnibus (GEO) database. Using the edgeR package to screen Differentially expressed genes (DEGs). Gene set enrichment analysis (GSEA) and gene set variation analysis (GSVA) were used to find pathways affecting VC. Gene ontology (GO) and Kyoto Encyclopedia of Genes and Genomes (KEGG) were performed on the DEGs. Meanwhile, using the String database and Cytoscape software to construct protein-protein interaction (PPI) networks and identify hub genes with the highest module scores. Correlation analysis was performed for hub genes. Receiver operating characteristic (ROC) curves, expression level analysis, GSEA, and subcellular localization were performed for each hub gene. Expression of hub genes in normal and calcified vascular tissues was verified by quantitative reverse transcription PCR (RT-qPCR) and immunohistochemistry (IHC) experiments. The hub gene-related miRNA-mRNA and TF-mRNA networks were constructed and functionally enriched for analysis. Finally, the DGIdb database was utilized to search for alternative drugs targeting VC hub genes.

Results

By comparing the genes with normal vessels, there were 64 DEGs in mildly calcified vessels and 650 DEGs in severely calcified vessels. Spp1, Sost, Col1a1, Fn1, and Ibsp were central in the progression of the entire VC by the MCODE plug-in. These hub genes are primarily enriched in ossification, extracellular matrix, and ECM-receptor interactions. Expression level results showed that Spp1, Sost, Ibsp, and Fn1 were significantly highly expressed in VC, and Col1a1 was incredibly low. RT-qPCR and IHC validation results were consistent with bioinformatic analysis. We found multiple pathways of hub genes acting in VC and identified 16 targeting drugs.

Conclusions

This study perfected the molecular regulatory mechanism of VC. Our results indicated that Spp1, Sost, Col1a1, Fn1, and Ibsp could be potential novel biomarkers for VC and promising therapeutic targets.

Introduction

Vascular calcification (VC) is a pathological process in which calcium phosphate accumulates in the vessel wall and is the basis for the development and progression of cardiovascular disease (Strauss et al., 2019). VC occurs mainly in the intima and intermediate layers of the arterial vasculature (Lee, Lee & Jeon, 2020). Intimal calcification is mainly atherosclerotic plaque caused by chronic inflammation of the vessel, leading to luminal narrowing and obstruction (Doherty et al., 2003; Durham et al., 2018). Intermediate calcification is mainly caused by the process of osteogenic differentiation mediated by vascular smooth muscle cells (VSMCs), leading to the loss of vascular elasticity (Chen, Zhao & Wu, 2020). VC increases with age, with chronic kidney disease, uremia, hypertension, and diabetes all contributing to this process (Giachelli, 2004; Mizobuchi, Towler & Slatopolsky, 2009). Studies have shown that VC is widespread in diabetic patients and that mortality increases with increasing calcification (Lindholt et al., 2020; Valenti et al., 2016). We found that calcium and phosphorus metabolism is disturbed in chronic kidney disease patients, which is a crucial factor contributing to the development of VC (London et al., 2003; Shroff, Long & Shanahan, 2013). VC leads to vascular stiffness and lumen narrowing, resulting in ventricular hypertrophy and heart failure (Jacob, 2003; Scialla et al., 2011). It has been reported that approximately 40% of deaths in people over 65 years of age are due to cardiovascular disease (Pescatore, Gamarra & Liberman, 2019). VC is a biomarker for the development of cardiovascular disease and peripheral vascular disease and is an important factor in the evaluation of health status in the elderly (Budoff et al., 2010).

In the past, VC has been considered a passive and unavoidable pathological process. However, recent findings suggest that the shift from the VSMCs phenotype to the osteoblast phenotype is a major reason for the occurrence of VC (Leopold, 2015). They concluded that VC is an active and regulated process similar to bone development and cartilage formation (Jiang et al., 2021; Towler et al., 2006). These results provide a solid theoretical basis for the study and treatment of VC. Although there have been many studies on the molecular mechanisms underlying the development of VC, effective preventive and therapeutic therapies are still lacking. Therefore, there is an urgent need to further search for hub genes and effective drugs for VC.

With the completion of the Human Genome Project, high-throughput detection and analysis technologies have been rapidly developed (Bell, 2004; Downes et al., 2019; Green, Watson & Collins, 2015). Bioinformatics analysis on the dataset from high-throughput sequencing has become one of the crucial methods in medical science research (Iqbal et al., 2021; Rung & Brazma, 2013; Vogelstein et al., 2013; Wheeler & Wang, 2013). The development of VC is a diverse and complicated process. We can study the biology of VC from more perspectives through genomics techniques. Analysis of high-throughput sequencing datasets can filter out many DEGs on VC. However, due to the complex molecular regulatory mechanisms of VC molecules, variability among different samples, and the various bioinformatics methods for analysis, some of the DEGs obtained may not be consistent each time. Therefore, to validate the accuracy of DEGs, we can use bioinformatics analysis methods from multiple VC datasets to explore common genes. The Fetuin-A, C5 and OPN protein have been reported to be valid and reliable as a bioinformatic marker of cardiovascular disease (Bostom et al., 2018; Kapetanios et al., 2016; Martínez-López et al., 2020; Van Campenhout & Golledge, 2009).

In this study, we used bioinformatics analysis to investigate the molecular regulatory mechanisms of hub genes that influence VC development and explore therapeutic agents that could target hub genes. We obtained DEGs between calcified and normal vessels from GSE146638, a gene expression profile downloaded from the GEO database. In addition, this dataset was performed with GSEA and GSVA to search for the biological pathways in VC. Immediately after, we performed GO and KEGG analysis on these DEGs. By constructing PPI networks, we screened for the highest scoring modules and hub genes. To validate that these hub genes can serve as markers of VC, we calculated each hub gene’s ROC curve area and expression levels in the uremia-induced VC and the normal groups. VC models were constructed to verify the expression levels of hub genes using RT-qPCR and immunohistochemistry. The pathways through which hub genes affect VC were explored using GSEA.

We then searched for miRNAs targeting regulated hub genes using the miRDB, miRWALK, and Targetscan databases and for target-regulated transcription factors using the KnockTF database. We constructed miRNA-mRNA and TF-mRNA networks and performed functional enrichment analysis on them. We identified 16 drugs that regulate hub genes affecting VC using the DGIdb database. This study will improve our understanding of the molecular mechanisms of VC and provide genomic-targeted therapy options for VC. And the process of our study is shown in Fig. 1.

Figure 1 Research flow chart.

Materials and Methods

Data acquisition and differential expression analysis

The RNA expression profile GSE146638 was obtained from the GEO database (https://www.ncbi.nlm.nih.gov/geo). It includes five normal samples, five samples with VC caused by uremia, and four samples with VC caused by vitamin D3. According to Jakob’s study, severe VC occurred in rats with uremia and slight VC in rats with vitamin D3 (Rukov et al., 2016). We respectively compared these two groups of samples with different degrees of VC with control samples. After obtaining the gene expression matrix, we used the edgeR package to look for DEGs between calcified and normal vessels using |log2FC| > 1 and FDR < 0.05 as cutoff criteria. To improve obtaining accurate and reliable DEGs, we used the VennDetail website to separate the intersection of the two groups’ upregulated and downregulated DEGs.

GSEA and GSVA

To find GO and KEGG entries affecting VC, we performed GSEA on normal and uremia-induced VC samples using the “clusterprofiler” R package, defining that pathways matching |NES| > 1, P-val < 0.05 and q-val < 0.05 were significantly enriched. Also, using “c2.cp.kegg.v7.4.entrez.gmt” and “c5.all.v7.4.entrez.gmt” as background gene sets, we performed a GSVA on normal and uremia-induced VC samples using the “GSVA” R package to explore further GO and KEGG entries affecting VC.

Functional enrichment analysis of DEGs

To better investigate the biological functions of DEGs, we performed functional enrichment analysis of vascular calcified DEGs using the “clusterprofiler” R package, which includes GO and KEGG pathways analysis, with P < 0.05 being statistically significant. GO analysis includes biological processes (BP), cellular components (CC), and molecular functions (MF).

PPI network and correlation analysis among hub genes

We constructed PPI networks using STRING (http://string-db.org) database and Cytoscape software for both sets of DEGs, and the significance threshold was set to an interaction score >0.4. The highest scoring modules were screened using the MCODE plugin of Cytoscape software, and the genes in the modules were defined as hub genes. We performed functional enrichment analysis of the hub genes in the highest-scoring modules using the Metascape (http://metascape.org) database. In addition, we obtained information of hub genes in VC and normal groups and explored the correlation between hub genes using the “corrplot” R package. GO annotation data were constructed from BP, CC, and MF levels using the “GOSemSim” R package to examine further whether hub genes are directly functionally similar. The geometric mean of these genes at the BP, CC, and MF levels was calculated to obtain a final score for a comprehensive assessment of the gene’s function in the three groups.

ROC curves and expression of hub genes

To evaluate the ability of these hub genes to distinguish calcified vessels from normal vessels, we extracted the expression profiles of hub genes in normal samples and uremia-induced VC samples. We plotted ROC curves for each hub gene using the “proc” R package. Meanwhile, we extracted the expression profiles of hub genes in normal samples and uremia-induced VC samples and visualized each hub gene’s expression level using the “ggplot2” R package.

Construction of miRNA-mRNA and TF-mRNA networks and functional enrichment analysis

Both miRNAs and transcription factors (TFs) can play a role in maintaining physiological stability by regulating the expression of target genes. We used miRWALK (http://mirwalk.umm.uni-heidelberg.de), miRDB (http://www.mirdb.org) and Targetscan (http://www.targetscan.org) databases to find miRNAs regulating Hub genes, taking the intersection of the results of these three databases. We used the KnockTF (http://www.licpathway.net/KnockTF/index.html) database to find transcription factors that regulate Hub genes. In addition, we visualized miRNA-mRNA and TF-mRNA networks with Cytoscape software and performed functional enrichment analysis of miRNA-mRNA networks and TFs-mRNA networks using the Metascape database.

Drug-gene interactions

We used the DGIdb (https://www.dgidb.org) database to retrieve the drugs targeting the hub genes of VC. Types of drug action include activation, inhibition and unknown.

GSEA and subcellular localization of each hub gene

To explore the function of each hub gene in VC, we divided each hub gene into high and low expression groups using the median expression level of each hub gene as the cut-off point. We performed GSEA on each hub gene using the “clusterprofiler” R package, setting the pathways meeting the criteria of |NES| > 1, P-val < 0.05, and q-val < 0.05 as significantly enriched. The localization of mRNA is an evolutionarily conserved phenomenon that controls many important biological processes, and the function of mRNA is closely related to its localization in the cell. To further understand the mechanism of hub genes involved in calcification, we obtained FASTA files of Spp1, Sost, Ibsp, Col1a1, and Fn1 from the Gene database and subsequently used the mRNALocater database to predict the subcellular localization of these genes.

Ethical statements and establishment of a rat model of VC

All the animal experiments were approved by the Ethics Committee of Guangxi Medical University (Approval Number: 201910030). Our studies conformed to the guidelines for the care and use of laboratory animals published by the Chinese Academy of Health Sciences. We obtained male Sprague Dawley rats weighing approximately 180 g from the Experimental Animal Center of Guangxi Medical University for the study. These rats were housed in a specific antigen-free environment and provided with adequate food and water.

We randomly divided the animals into two groups of 10 animals each. The VC model was established by intraperitoneal injection of vitamin D3 300,000 IU/kg once daily for 14 days. To ensure the univariate principle of the experiment, the negative control group was injected with an equal amount of saline. The rats were treated with the cervical dislocation method at the end of the experiment, and then the aorta was collected.

RT-qPCR experiments of aorta

To measure the expression of Spp1, Sost, Ibsp, Col1a1, and Fn1 in calcified vessels, we performed reverse transcription and real-time fluorescence polymerase chain reaction on RNA. The primers used are shown in Table 1. Spp1, Sost, Ibsp, Col1a1 and Fn1 expressions were normalized by β-actin, with the relative expressions calculated by comparison with the 2−ΔΔCT method.

Table 1 RT-qPCR primers for genes.

Gene	Primer sequence	
IBSP	Forward	5′-CACTGGAGCCAATGCAGAAGA-3′	
Reverse	5′-TGGTGGGGTTGTAGGTTCAAA-3′	
SOST	Forward	5′-CAGCTGTTTTTGTCTGCCCC-3′	
Reverse	5′-GTCATCCACCTGCAGAGCTT-3′	
FN1	Forward	5′-CCACAGCCATTCCTGCGCCA-3′	
Reverse	5′-TCACCCGCACTCGGTAGCCA-3′	
COL1A1	Forward	5′-GAGGGCCAAGACGAAGACATC-3′	
Reverse	5′-CAGATCACGTCATCGCACAAC-3′	
SPP1	Forward	5′-GAAGTTTCGCAGACCTGACAT-3′	
Reverse	5′-GTATGCACCATTCAACTCCTCG-3′	
β-Actin	Forward	5′-GGAGATTACTGCCCTGGCTCCTA-3′	
Reverse	5′-GACTCATCGTACTCCTGCTTGCTG-3′	

Alizarin red staining, Von kossa staining and immunohistochemical experiments of aorta

We fixed the thoracic aorta in 4% paraformaldehyde and embedded it in a wax block. We cut the wax block into 5 um sections, and the sections were dewaxed with xylene and dehydrated with the alcohol solution. For Alizarin red staining, we placed the sections in Alizarin red staining solution for 5 min and then photographed them under a light microscope. For Von Kossa staining, we incubated the sections in 1% silver nitrate in sunlight for 1 h followed by 2% sodium thiosulfate for 2 min, and then photographed with a light microscope. Positively stained areas were detected using ImageJ analysis. For immunohistochemical staining of thoracic aorta, we removed paraffin from the sections, dehydrated them in graded alcohol solution, closed them with 5% fetal bovine serum for 1 h, and stained them with Spp1 (22952-1-AP 1:100), Sost (21933-1-AP 1:100), Ibsp (ABP60403 1:200), Col1a1 (67288-1-Ig 1:2,500) and Fn1 (66042-1-Ig 1:500) primary antibodies were soaked overnight at 4 °C. The following day, we incubated the sections with the corresponding secondary antibodies for 1 h at room temperature. According to the manufacturer’s requirements, we first stained with the DAB kit, repeated the staining with hematoxylin, and then photographed with a light microscope. ImageJ was used to analyze the positively stained areas.

Statistical analysis

The data set was analyzed using Rstudio (version 4.0.2). Differential expression analysis was performed using t-test to determine P-values and FDR. All results are expressed as mean ± SD. We performed statistical analysis of the results by Student’s t-test using GraphPad Prism software. P < 0.05 was considered statistically significant.

Results

Identification of DEGs in VC

Comparing the genes of uremia-induced VC with normal samples in GSE146638, we found 650 DEGs, including 405 up-regulated and 245 down-regulated genes (Fig. 2A). Comparing the genes of vitamin D3-induced VC with those of normal vessels in GSE146638, we found 64 DEGS, including 42 up-regulated and 22 down-regulated genes (Fig. 2B). We then intersected the results of the two groups of up-and down-regulated DEGs separately to obtain a set of DEGs containing five down-regulated genes (Fig. 2C) and nine up-regulated genes (Fig. 2D).

Figure 2 Screening of DEGs.

(A) Volcano plot of DEGs between uremia-induced vascular calcified samples and normal control samples. (B) Volcano plot of DEGs between vitamin D3-induced vascular calcified samples and normal control samples. Red dots: genes significantly up-regulated in VC. Gray-green dots: genes significantly down-regulated in VC; black dots: genes not differentially expressed. |log2FC| > 1 and FDR < 0.05 as cutoff criteria. (C) The Venn diagram of down-regulated DEGs with different degrees of vascular calcification. (D) The Venn diagram of up-regulated DEGs with different degrees of VC.

The result of GSEA and GSVA

GO and KEGG analysis of genes between uremia-induced VC and normal groups were performed by using the GSEA method. In GO BP analysis, genes were primarily rich in antigen processing and peptide antigen presentation, positive regulation of leukocyte cell adhesion and cell activation (Fig. 3A). In GO CC analysis, genes were mainly rich in MHC protein complexes, mitochondrial respiratory chain complex I (Fig. 3B). In GO MF analysis, genes were mainly rich in glycosaminoglycan binding, heparin-binding and receptor-ligand activity (Fig. 3C). In KEGG pathway analysis, genes were rich in allograft rejection, reactive oxygen species separation, and ECM-receptor (Fig. 3D). Also, we performed GSVA on genes from uremia-induced VC and normal samples. In GO analysis, genes from VC samples were mainly rich in cytoskeleton organization, phosphoprotein binding, and Transporter vesicles (Fig. 4A). In KEGG analysis, genes were rich in Koinuma targets of smad2 or smad3 and Gobert oligodendrocyte differentiation (Fig. 4B).

Figure 3 The GESE between the uremia-induced VC group and the normal group.

The GSEA includes GO BP analysis (A), GO CC analysis (B), GO MF analysis (C), and KEGG pathway analysis (D).

Figure 4 The GSVA between the uremia-induced VC group and the normal group.

The GSVA includes GO analysis (A) and KEGG pathway analysis (B).

Functional enrichment analysis and pathway of DEGs

We performed GO and KEGG pathway analysis of DEGs in the uremia-induced VC group and normal controls. In GO BP analysis, we found that DEGs were rich in ossification and response to glucocorticoid (Fig. 5A). In GO CC analysis, DEGs were rich in lysosomes, cell vacuoles, and extracellular matrix (Fig. 5B). In GO MF analysis, DEGs were mainly rich in amide binding, cell adhesion molecule binding, and sulfur compound binding (Fig. 5C). In KEGG pathway analysis, DEGs were primarily rich in human T-cell leukemia virus one infection, focal adhesion and lysosomes (Fig. 5D).

Figure 5 Functional enrichment analysis and pathway of DEGs.

(A) GO BP analysis. (B) GO CC analysis. (C) GO MF analysis. (D) KEGG pathway analysis.

PPI network construction, hub genes identification and correlation between hub genes

We constructed and analyzed PPI networks for two groups of DEGs with different degrees of VC using the STRING database and Cytoscape software. In the uremia-induced VC group, 571 DEGs with a score >0.4 were selected for PPI network construction (Fig. 6A). In the vitamin D3-induced VC group, 36 DEGs with scores >0.4 were selected for PPI network construction (Fig. 6B). We also constructed a PPI network for the intersection of DEGs with two different degrees of VC. And the most significant module consisting of five genes and 18 edges was found using the MCODE plug-in of Cytoscape software (Fig. 6C). Next, we performed functional enrichment analysis of these hub genes using the Metascape database (Fig. 6D). In GO BP analysis, we found that hub genes were rich in ossification, response to steroid hormones, and response to mechanical stimulation. In GO CC analysis, hub genes were rich in the extracellular matrix. In GO MF analysis, hub genes were rich in cell adhesion molecule binding, and integrin binding. And in KEGG pathway analysis, hub genes were rich in ECM-receptor interaction. In addition, correlation analysis of hub genes (Spp1, Sost, Col1a1, Fn1, Ibsp) was highly correlated (Fig. 6E). Also, these five hub genes were functionally related, with IBSP occupying the most significant position (Fig. 6F).

Figure 6 PPI network construction, hub genes functional enrichment analysis and correlation.

PPI network construction, hub genes functional enrichment analysis and correlation. (A) The PPI network of DEGs in the uremic-induced VC group. (B) The PPI network of DEGs in the vitamin D3-induced VC group. (C) The PPI network of the intersection of two DEGs with different degrees of calcification. (D) Functional enrichment analysis and pathway analysis of hub genes. (E) The correlations between hub genes. (F) The functional similarities between hub genes.

ROC curves and expression of hub genes

We constructed ROC curves of each hub gene separately and found that their area under ROC curves of Sost, Ibsp, Fn1, Col1a1, and Spp1 were all close to one in the uremia-induced VC group in GSE146638, with the normal group as control (Fig. 7A). These indicate that these five genes have a good ability to discriminate between calcified and normal vessels. Meanwhile, by verifying the expression of these hub genes in the uremia-induced VC group (Fig. 7B), we found that Sost, Ibsp, Fn1, and Spp1 were highly expressed in calcified samples, while Col1a1 was lowly expressed.

Figure 7 ROC curves and expression of hub genes in the dataset.

(A) The ROC curve of each hub gene in uremia-induced VC samples was compared with normal vessels. (B) The expression of each hub gene in uremia-induced VC samples was compared with normal vessels.

Validation of hub gene expression in rat aorta with VC

By performing Alizarin red staining and Von kossa staining on the aorta, we found significant calcium salt deposition on the aortic wall of rats induced with vitamin D3, which indicated that the constructed model of vascular calcification was successful (Fig. 8A). We demonstrated by RT-qPCR experiments that Sost, Ibsp, Fn1, and Spp1 mRNA expression were highly expressed in the calcified aorta, while Col1a1 mRNA was lowly expressed (Fig. 8B). Also, we demonstrated by immunohistochemistry experiments that Sost, Ibsp, Fn1, and Spp1 protein expression were highly expressed in the calcified aorta, while Col1a1 protein was lowly expressed (Fig. 8C).

Figure 8 Validation of each hub gene expression in rat aorta with VC.

(A) Alizarin red staining and Von Kossa staining of rat aorta. The positively stained regions were analyzed using ImageJ. The scale bar represents 500 um (black) and 200 um (red). (B) Relative expression in RT-qPCR of these hub genes. (C) The expression of hub genes in the thoracic artery were detected by immunohistochemistry and the average optical density values were analyzed by ImageJ. The scale bar represents 500 μm (black) and 100 μm (green).

GSEA and subcellular localization of hub genes

By performing GSEA analysis of each hub gene, we found that Ibsp, Sost, Fn1, and Spp1 were primarily rich in allograft rejection, ECM -receptor interaction and phagosome. And Col1a1 were primarily rich in citrate cycle and oxidative phosphorylation (Figs. 9A–9E). We found that Sost and Col1a1 were mainly distributed in the cytoplasm, and Ibsp, Spp1 and Fn1 were primarily distributed in the nucleus (Fig. 9F). They play similar roles although they are mainly distributed in different locations of the cell.

Figure 9 GSEA and Subcellular localization of each hub gene.

(A) GSEA of Ibsp. (B) GSEA of Sost. (C) GSEA of Fn1. (D) GSEA of Col1a1. (E) GSEA of Spp1. (F) Subcellular localization of each hub gene.

Construction and functional enrichment analysis of miRNA-mRNA networks and TF-mRNA networks

We searched for target-regulated hub gene miRNAs using miRWALK, miRDB, and Targetscan databases and then used the results of these databases as intersections. By constructing miRNA-mRNA networks, we found 109 miRNAs targeting regulation of Col1a1, 94 miRNAs targeting regulation of Fn1, 25 miRNAs targeting regulation of Ibsp, 49 miRNAs targeting regulation of Sost, and Spp1 regulated by 18 miRNAs (Fig. 10A). Meanwhile, we found that miRNA-mRNA network functional enrichment analysis primarily focused on gene silencing, ossification, and ECM-receptor interactions of miRNAs (Fig. 10B). We used the KnockTF database to search for target-regulated hub gene transcription factors. We found 80 TFs regulating Col1a1, 45 TFs regulating Fn1, 48 TFs regulating Ibsp, 38 TFs regulating Sost, and 37 TFs regulating Spp1 (Fig. 11A). Meanwhile, we found that TF-mRNA network functional enrichment analysis mainly focused on mRNA metabolic processes, myeloid cell differentiation, stem cell differentiation, and transcriptional regulator complexes (Fig. 11B).

Figure 10 miRNA-mRNA network construction and functional enrichment analysis of hub genes.

(A) miRNA-mRNA network construction. (B) Functional enrichment analysis.

Figure 11 TF-mRNA network construction and functional enrichment analysis of hub genes.

(A) TF-mRNA network construction. (B) Functional enrichment analysis.

Drug‑gene interaction

We identified 16 drugs that target these hub genes using the DGIdb database (Fig. 12). We found that Blosozumab, Romosozumab, and Setrusumab act on Sost, Daidzein and Insulin act on Ibsp, and L19il2, L19sip131l, L19tnfa, and Ocriplasmin act on Fn1, Ocriplasmin and Collagenase clostridium histolyticum act on Col1a1, Tacrolimus, Calcitonin, Alteplase, Gentamicin, Ask-8007, and Wortmannin all act on Spp1. These offer the possibility of genetic treatment of VC.

Figure 12 Drug‑gene interaction.

The drugs targeting hub genes were obtained by using the DGIdb database. The black line represents inhibited interactions. The gray line represents unknown interactions.

Discussion

VC is a normal part of aging, but diabetic nephropathy and chronic kidney disease, among others, can exacerbate this pathological process. Failure of early screening and diagnosis of VC leads to progressive worsening of the lesion. The degree of VC seriously affects the recovery from cardiovascular system diseases and is one of the vital indicators to evaluate the health status of the elderly. Therefore, the study of biomarkers of VC is significant for the early diagnosis and treatment of the disease. With the development of high-throughput sequencing technology, it has started to be widely applied to find the hub genes of diseases. The re-analysis of sequencing data provides a convenient and comprehensive platform to reveal the molecular mechanism of VC occurrence and find effective target drugs for the treatment of VC.

In this study, we analyzed the VC expression profile GSE146638 screened from the GEO database. It includes five normal samples, five samples with severe VC caused by uremia, and four samples with mild VC caused by vitamin D3. Compared to normal controls, we found 64 DEGs (including 42 up-regulated and 22 down-regulated) in the vitamin D3-induced VC group and 650 DEGs (including 405 up-regulated and 245 down-regulated) in the uremia-induced VC group. We performed GO and KEGG analysis on DEGs. In GO analysis, we found that DEGs were rich in ossification, lysosomes, and extracellular matrix. In KEGG pathway analysis, DEGs were primarily rich in Staphylococcus aureus infection, FoxO signaling pathway, Human T-cell leukemia virus one infection. Focal adhesion and Lysosomes. Some of these pathways are involved in the process of VC. Intrinsic stiffness and extracellular matrix remodeling of VSMC contribute to osteogenic differentiation and calcification of VSMC, and ossification and calcification are closely related (Chen, Zhao & Wu, 2020). Disruption of extracellular matrix integrity can attenuate medial aortic calcification in its early stages (Uto et al., 2021). Upregulation of FOXO1/3 inhibits VC (Deng et al., 2015). Mitral annular calcification (MAC) may act as a nidus for infection, especially with Staphylococcus aureus infection (Pressman et al., 2017).

We constructed a PPI network for these DEGs and screened for a top-scoring gene module (Spp1, Sost, Col1a1, Fn1, Ibsp). Meanwhile, we found that these module genes were involved in the whole process of VC progression. We performed the functional enrichment analysis of these hub genes. In GO BP analysis, hub genes were mainly rich in ossification, response to mechanical stimulation, and vascular development. In GO CC analysis, hub genes were rich in the extracellular matrix. In GO MF analysis, hub genes were rich in cell adhesion molecule binding, and integrin binding. And in KEGG pathway analysis, hub genes were enriched in ECM receptor interaction. Numerous studies have shown that VC is mainly produced by the process of osteogenic differentiation mediated by VSMCs (Massy et al., 2008; Shroff et al., 2008). Hypothyroidism and vitamin D deficiency are associated with various cardiovascular diseases, including hypertension, atherosclerosis, VC, and renal failure (Khundmiri, Murray & Lederer, 2016; Pilz et al., 2016). Matrix GLA proteins have a huge role in maintaining arterial vascular stability, and the lack of matrix GLA promotes VC (El-Maadawy et al., 2003; Luo et al., 1997). These results indicate that our approach to studying VC by bioinformatics is accurate and effective. We found that these hub genes have a strong ability to distinguish whether they are VC or not. In the VC expression profile GSE146638, we found that the expression of Ibsp, Sost, Fn1 and Spp1 was high in samples with VC, while the expression of Col1a1 was low in samples with VC. In addition, our results from RT-qPCR and immunohistochemistry experiments were consistent with those in the dataset. Previous studies have shown that high expression of Fn1 is found in arterially calcified aorta and may induce VC through the ERK pathway (Ding et al., 2006). Spp1 may prevent high phosphorus-induced renal calcification and VC (Paloian, Leaf & Giachelli, 2016). Sost prevents VC through a negative feedback mechanism (Catalano et al., 2020; De Maré et al., 2019). We found that Ibsp is highly expressed in human calcified plaques (Ayari & Bricca, 2012). Col1a1 has been less studied in VC, and the exact molecular regulatory mechanism remains unclear. It has been shown that activation of parathyroid hormone receptors inhibits Col1a1 expression and attenuates aortic fibrosis (Cheng et al., 2010).

To further search for mechanisms of action that affect VC, we used GSVA and GSEA to search for pathways associated with VC. The use of GSEA identified several pathways that affect VC, such as ECM-receptor interaction and Hematopoietic cell lineage. Alterations in the extracellular matrix are associated with atherosclerosis, restenosis, and heart failure (Chistiakov, Sobenin & Orekhov, 2013). Extracellular matrix (ECM) proteins are the first component contributing to atherosclerosis (Lacolley et al., 2017). Post-translational modification of PAR initiates biomineralization by concentrating calcium into spherical structures that can form apatite minerals on the ECM (Duer, Cobb & Shanahan, 2020). Platelets represent the cellular interface between hemostasis, inflammation, and atherosclerosis and have myeloid precursors that promote atherosclerosis (Foresta et al., 2013). Also, by GSVA analysis, we identified multiple pathways affecting VC such as cytoskeleton organization, and apoptotic process. Small contractions of the cytoskeleton may attach integrins to sites on matrix proteins. This results in a sensation of stiffness, which may be caused by the release of differentiation factors through the deformation of binding proteins (Hsu et al., 2016). Apoptosis is a factor that promotes vascular calcification in patients with chronic kidney disease (CKD), and MitoQ inhibits vascular calcification by suppressing apoptosis of vascular smooth muscle cells (VSMC) via the Keap1/Nrf2 pathway (Cui et al., 2020).

The complexity and diversity of VC have hindered our accurate understanding of the disease and the discovery of drugs with precise targets. In this study, we used a comprehensive bioinformatics approach to find hub genes that influence the progression of VC and the drugs that target hub genes. Although some of hub genes have been studied, the molecular mechanisms behind VC remain unclear. Therefore, we predicted miRNAs and transcription factors that regulate central genes and constructed miRNA-mRNA and TF-mRNA networks, respectively. Some of these miRNAs and transcription factors have been shown to play an integral role in the pathology of vascular calcification. miR-29b-3p attenuates vascular calcification by regulating matrix metalloproteinase 2 (Jiang et al., 2017). miR-25 protects glucocorticoid-induced apoptotic process in VSMCs thereby reducing vascular calcification (Zhang et al., 2019). miR-134-5p exacerbates vascular calcification by regulating the expression of histone deacetylase 5 in high phosphate conditions (Choe et al., 2020b). CircSmoc1-2 attenuates vascular calcification by regulating the miR-874-3p/Adam19 axis (Ryu et al., 2022). miR-26-5p affects the pathological process of VC in patients with chronic kidney disease by regulating PTEN expression (Chang et al., 2020). miR-27a-3p attenuates vascular calcification by regulating activating transcription factor 3 (an osteogenic transcription factor) (Choe et al., 2020a). Krupp-like factors 4 is a zinc-finger transcription factor and synergizes with Runx2 to enhance arterial calcification (Yoshida, Yamashita & Hayashi, 2012). SOX transcription factor affects endothelial-mesenchymal transition leading to vascular calcification (Yao, Yao & Boström, 2019). Deacetylation of Sp1 reduced the binding of Sp1 to the BMP2 promoter, thereby reducing apoptosis and calcium deposition (Zhang et al., 2021). The transcription factor XBP1, synergistically with Runx2, is involved in VC due to oxidative stress (Liberman et al., 2011). TEAD4 is a nuclear transcription factor that inhibits vascular calcification by regulating the Wnt/β-Catenin signaling pathway (Cong et al., 2021). By intervening in VC hub genes, it may be a new direction for individualized genomic therapy of VC.

Vascular calcification can be caused by various diseases, but there is no effective treatment to reduce or reverse vascular calcification. Current therapy of VC is primarily through correction of mineral metabolism markers such as calcium, vitamin D, parathyroid hormone (PTH), and phosphorus (Singh, Tandon & Tandon, 2021). Clinical investigations of drugs used to treat VC include phosphate binders, calcium mimetics, bisphosphonates, vitamin D, and vitamin K. Bisphosphonates inhibit nucleation and growth of hydroxyapatite and have been used to treat osteoporosis (Lomashvili et al., 2009). In uremic animals, pamidronate and etidronate inhibit vascular calcification independent of bone resorption (Giger, Castagner & Leroux, 2013). Vitamin K promotes blood clotting and calcification homeostasis (Shea & Booth, 2019). Sodium thiosulfate chelates calcium to form highly soluble complexes, thereby inhibiting VC (Schlieper et al., 2009). IP6 hinders VC by inhibiting hydroxyapatite microcrystals II without directly affecting calcium and phosphorus blood levels (Ferrer et al., 2018). TNAP is a tissue non-specific alkaline phosphatase (ALP) isoenzyme that provides a site for bone mineralization by inactivating pyrophosphate and upregulating phosphate ion levels. Inhibitors targeting TNAP are under investigation (Singh, Tandon & Tandon, 2021).

Current treatments for VC are focused on maintaining calcium and phosphorus homeostasis. Traditionally drug development has relied on physical experiments and drug compounds, but adding indications to existing drugs to treat other diseases is a valid alternative. For example, sildenafil (Viagra) was initially used to treat angina but is now used to treat erectile dysfunction. Using the Dgidb database, we screened 16 drugs targeting these central genes, some for the treatment of VC and some for other diseases. Insulin has been reported to attenuate calcification of vascular smooth muscle sarcomeres (Wang et al., 2007). Romosozumab is used for the treatment of osteoporosis in women (Turk et al., 2020). l19-IL2 selectively activates Tregs to eliminate local inflammation and induce plaque shrinkage in mice, thereby inhibiting atherosclerotic plaques (Dietrich et al., 2012). The results of probability score selection suggest that daidzein as an antioxidant may attenuate VC (Chao et al., 2022). Wortmannin may treat VC by inhibiting phosphatidylinositol 3-kinase-related DDR1 (Lino et al., 2018). Tacrolimus immunosuppression has a therapeutic effect on bone loss and the tendency of cyclosporine A to cause vascular disease (Hofbauer et al., 2001). Immune, antioxidant, and phosphatidylinositol 3-kinase inhibition pathways may be investigated as a treatment for VC, complementing the traditional therapeutic approach of maintaining calcium and phosphorus homeostasis. However, more basic research and clinical trials are needed in the future to verify how effective these drugs are in treating VC and whether they cause side effects.

However, there are still some limitations of our study. First, due to the lack of other vascular calcification datasets, it was impossible to validate our analysis results using different datasets. Second, although we constructed a vascular calcification model to validate the expression levels of hub genes, the validity of the final analysis results needs to be verified in human tissues due to species differences. Third, although we have used bioinformatics methods to preliminarily analyze the gene sets involved in VC, the connection of these gene sets with actual biology needs further experimental proof.

Conclusions

In conclusion, through bioinformatics analysis and experimental validation, we found that these hub genes (Spp1, Sost, Col1a1, Fn1, Ibsp) are consistently present and play a regulatory role throughout the progression of VC. These hub genes can act as biomarkers for the diagnosis and treatment of VC.

Supplemental Information

Supplemental Information 1 Raw data of PCR results.

Click here for additional data file.

Supplemental Information 2 GSE146638.

Click here for additional data file.

Supplemental Information 3 Author Checklist.

Click here for additional data file.

Additional Information and Declarations

Competing Interests

Author Contributions

Data Availability

The authors declare that they have no competing interests.

Chuanzhen Chen conceived and designed the experiments, performed the experiments, analyzed the data, prepared figures and/or tables, authored or reviewed drafts of the paper, and approved the final draft.

Yinteng Wu conceived and designed the experiments, analyzed the data, authored or reviewed drafts of the paper, and approved the final draft.

Hai-lin Lu conceived and designed the experiments, analyzed the data, authored or reviewed drafts of the paper, and approved the final draft.

Kai Liu conceived and designed the experiments, analyzed the data, authored or reviewed drafts of the paper, and approved the final draft.

Xiao Qin conceived and designed the experiments, performed the experiments, analyzed the data, prepared figures and/or tables, authored or reviewed drafts of the paper, and approved the final draft.

The following information was supplied regarding data availability:

The raw measurements are available in the Supplemental File.

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
