# Peer review of "Identification of potential biomarkers of vascular calcification using bioinformatics analysis and validation in vivo"

_PeerJ, doi:10.7717/peerj.13138_

## Round 0.1 · original submission · Major Revisions

Consider the reviewers' comments to improve the quality of the manuscript.

Reviewer 1 ·

Basic reporting

The article entitled “Identification of potential biomarkers of vascular calcification using bioinformatics analysis and validation in vivo” by Chen et al. has demonstrated the potential prognostic biomarker and promising therapeutic targets for vascular calcification (VC) using bioinformatics analysis based on RNAseq data from GEO. In addition, the expression levels of candidate hub genes were verified in mouse tissues. They finally concluded that these genes might be a potential prognostic biomarker and promising therapeutic targets for VC. However, several improvements can be made to the manuscript to improve basic reporting and overall usefulness.

Experimental design

In order to strengthen the manuscript, I suggest the following experiments:
- The authors have used only a small RNAseq dataset from GEO. Please explain why you used this dataset despite the accessibility of many more RNAseq and microarray data from other public databases.
- To confirm the reliability of the analyzed results, the expression levels of all candidate genes in VC using different datasets and different databases have to be further examined.
- The immunohistochemistry or ELISA in human samples must be used to confirm the bioinformatics finding.

Validity of the findings

no comment

Additional comments

Overall, the manuscript has been written in a coherent manner. However, the author needs to edit/rewrite some part of the manuscript. In addition, the authors need to perform several extra experiments.

Reviewer 2 ·

Basic reporting

Overall, the manuscript was written using clear and unambiguous professional English throughout. Citations were used correctly. The article structure complies with peerJ and raw data are provided.

Minor point: please check some suggestions provided in the attached PDF, regarding subtitle formating and recommended subtite change.

Experimental design

The manuscript attempts to address the lack of biomarkers for vacular calcification. By using public data from GEO dataset together with self-generated data, this scientific question was answered, with 5 key hub genes identified as the central biomakers associated with VC progression.

Methods were discussed adequately, although the details regarding the source of data for each section in the result section can be further revised to improve clarity.

Validity of the findings

In general, all claims have been backed up with evidence although some figures were incorrectly cited (refer to attached PDF, specifically figure 7A and B).

Specific points
1) Regarding GSEA analysis, the authors are recommended to use NES score to compare the significance between the different hallmark gene sets

2) On the reporting of the 16 potential drugs, the findings do not address how each drug will affect the expression of these 5 hub genes. To serve as therapy for VC, the regimen must function, to the minumum, in the opposite direction from the change in expression between VC model and the control group. Authors are recommended to apply connectivity mapping together with the clue.io database for this analysis.

3) This analysis were performed on rat model. The authors are advised to discuss how consistent these results will be when performing in human specimens.

4) Specifically for the miRNA-transcriptional factors-mRNA analysis, it is very challenging to determine the validity of these results, and how they are unique from other diseases.

Additional comments

The result section can perhaps be revised to address which of the dataset were used for each analysis. In addition to the discovery of the 5 hub genes, the authors are advised to discuss how the identified GO terms from GSEA and KEGG are relevant to the VC progression. Currently, some of the GO terms were not easily relatable to the VC progression.

Annotated reviews are not available for download in order to protect the identity of reviewers who chose to remain anonymous.

·

Basic reporting

The English is good but it can be better with English editors.

Experimental design

1. Because you used the genes from public dataset (GSE146638), so please contrast how different the previous study with your finding.
2. Is there any public VC dataset performed experiment in human? In the present manuscript relied only in rat (both public data and the animal model).
3. In my opinion, the |log2FC| > 1 is too soft cut off. I would like to see the stricter cut off result.
4. How to make sure that the rat in the experiments got VC?
5. The number of passed genes from each step and the number of gene used in each experiment are not clear. I suggest to rewrite the experimental design in the Method section.

Validity of the findings

1. Add more discussion on the drugs that routinely used and compare to the presented drug in the manuscript.
2. In figure 2, why the cut point for Fig2A is different from that of fig2B.
3. Please correct the word GESE to GSEA in the legend of figure 3.

Additional comments

The manuscript by Chen C. et al showed very interesting results of genes associated with vascular calcification (VC) and the possible targets for treatments. I really appreciate the experimental design which combines bioinformatics from public data and the animal experiment. However, I have some questions and comments on the manuscript.

---

## Round 0.2 · Minor Revisions

The manuscript has been improved but a few minor suggestions of Reviewer 2 remain to be addressed

Reviewer 2 ·

Basic reporting

The authors have made writing improvement to address issues on validity of the study using rat instead of human, by stating that further experiment on human cell model is still necessary. Previously identified annotations were corrected. Literature references were corrected cited. Raw data were shared.

Experimental design

Experimental method sections were adequate to follow and repeat the work.

Validity of the findings

1 Regarding the connections of the identified gene ontology terms on VC progression, the authors attempted to justify the connections of some of the identified gene sets by referring to prior literature. While this sounds plausible, we cannot really be certain on the connection of these gene sets to actual biology, apart from serving as hypothetical inference.
2 Like what was discussed in the previous review process, the identified 16 drugs were taken from DGIdb database, but there is no evidence that these drugs will be candidate therapeutic agents for VC at all in this study. Providing the structures of these compounds alone show no connection to their ability to serve as therapeutic agents, but may somewhat mislead the audience in understanding that they are drug candidates. The authors are recommended to justify how these 16 drugs are linked to VC progression. Statement like '...enriching the options for precisely targeted therapy of VC. Our study can provide some help for targeted genomic therapy for VC in the future.' in the conclusion holds no concrete evidence from this study to support this claim yet.

Additional comments

From the rebuttal letter, authors claimed that the NES score was already performed and gave a similar result to the old approach, and claimed that there is no need to alter the picture. Without seeing this actual result, we cannot confirm this claim. In fact, the authors added statement in the Methods section about the NES score, but did not include this result at all.

Based on the response from the authors regarding their miRNA and transcription factors association, it is still impossible for us to determine the validity of these findings and how they are different from other diseases. The authors referred to 'numerous articles using similar approach' but the former question was asked about how to connect this result to VC progression, and its validity.

·

Basic reporting

The manuscript is much better than the previous one.

Experimental design

no comment

Validity of the findings

no comment

Additional comments

I appreciate the improvement of the current manuscript. The authors answered and corrected the manuscript as suggested.

---

## Round 0.3 · accepted · Accept

The authors addressed properly reviewer comments.